# Structural Changes in the Skeletal Muscle of Pigs after Long-Term Administration of Testosterone, Nandrolone and a Combination of the Two

**DOI:** 10.3390/ani13132141

**Published:** 2023-06-28

**Authors:** Kristýna Skoupá, Andrej Bátik, Kamil Št’astný, Zbyšek Sládek

**Affiliations:** 1Department of Animal Morphology, Physiology and Genetics, Faculty of AgrSciences, Mendel University in Brno, Zemedelska 1, 613 00 Brno, Czech Republic; kristyna.skoupa@mendelu.cz (K.S.); andrej.batik@gmail.com (A.B.); 2Veterinary Research Institute in Brno, Hudcova 296/70, 621 00 Brno, Czech Republic; kamil.stastny@vri.cz

**Keywords:** anabolic steroids, histology, muscle fibers, satellite cells, pigs

## Abstract

**Simple Summary:**

Anabolic steroids (AASs) are used for more efficient fattening of animals all over the world, with the exception of some countries, including the European Union. There is a strict ban on the use of these substances, and protection and detection methods also depend on this, but in recent years, they have reached their limits. New methods of detection and obtaining detailed information on the actions of AASs are now required. This study focuses on histological changes in the skeletal muscle of pigs, one of the most consumed animals in the EU. The aim was to determine the dependence of the histological parameters of pig muscle on the application of AASs and to supplement the information on these interactions that is missing in pig models.

**Abstract:**

Anabolic steroid hormones (AASs) are used in most countries of the world to accelerate the growth of animals, increase the volume of their muscles and thereby increase meat production. However, there is a strict ban on the use of AASs in the fattening of all animals in all countries of the European Union, and there must therefore be effective methods of detection and control of these substances. Methods based on chromatography and mass spectrometry may no longer be completely effective when faced with new synthetic steroids of unknown chemical structures and low concentrations. Therefore, there is an effort to develop new methods of AAS detection, based primarily on the monitoring of biological changes at the level of gene expression or changes in metabolism or structure at the cellular level. More detailed knowledge of the mechanisms of action of AASs on tissues is essential for these methods, and histological changes are one of them. In this study, we report histological changes in muscle structure after AAS application, specifically in the size of muscle fibers, the amount of endomysium and the number of nuclei and satellite cells in muscle fibers. A pig model was also intentionally used for the study, as no such study has been carried out on this species, and at the same time, pork is one of the most consumed meats across Europe. The results of histology and fluorescent antibody labeling showed that AASs increased the diameter and surface area of muscle fibers and also significantly increased the number of satellite cells on the fiber surface. The evident correlations between the number of satellite cells, all nuclei and the diameters of muscle fibers between some experimental groups provide evidence that the selected histological parameters could be additional detection mechanisms for screening a large number of samples and indicate the possibility of the presence of AASs in pork meat in the future.

## 1. Introduction

Steroid hormones are naturally occurring hormones in the body, synthesized from the precursor molecule cholesterol in the endocrine cells of the adrenal cortex, the Leydig cells of the testes and the follicular cells of the ovaries and placenta. They control sexual development and reproduction and participate in the regulation of the metabolism of carbohydrates, minerals and especially proteins. The effects associated with protein synthesis, muscle growth and masculinization are generally referred to as anabolic. These properties of steroid hormones have been used to create synthetic substances to promote rapid muscle growth, known as anabolic-androgenic steroids (AASs) [1,2,3]. AASs are chemically synthetic derivatives of the male sex hormone testosterone and/or its structural modifications. These modifications can influence the ratio of anabolic and androgenic properties of the steroid and its stability in the body. However, there is still no AAS available that has only anabolic or only androgenic features, but some synthetic steroids show a partial dissociation between these two activities [4,5,6].

AASs are often used in the fattening of meat-producing animals throughout the world, with the exception of EU member states [7], Brazil [8] and some countries in Oceania [9], where their use is strictly prohibited by law. Most states approve the use of a number of anabolic-based preparations for fattening cattle and sheep, but no type of AAS is approved for growth acceleration in dairy cows, pigs and poultry [6]. Globally, it is estimated that up to 90% of cattle in fattening outside the EU are treated with at least one type of anabolic growth promoter [3]. Their levels usually have strict limits. However, due to the ability of AASs to greatly accelerate muscle growth and, thereby, speed up and fatten animals and increase the profit from the meat sold, AASs are abused illegally, in more than permitted doses and in unauthorized combinations. But proving this abuse is very challenging. One factor is the presence of natural endogenous steroid hormones, the physiological concentrations of which differ depending on the conditions, age and sex of the animal and the very conditions of analysis and statistical methods of different laboratories. A second is the use of newly invented steroid substances and their mixtures, which have unknown chemical structures and are not included in the evaluation panels of control laboratories. Most of these synthetic anabolics are ester versions of potentially endogenous steroids; however, demonstrating the presence of a steroid ester as evidence of abuse is not always possible because the ester is already metabolized by the time it reaches the test matrix. Therefore, some type of quantitative multivariate threshold approach is usually required to confirm abuse [10].

The increasingly frequent abuse of steroid preparations is mainly supported by their very easy availability on the market (and black market) as well as their low price. The Official Medicines Control Laboratories (OMCLs) report, which describes the situation of the anabolic steroid market in the EU, reports that only 0.06% of seized AASs were obtained legally, and in total, more than 80 different undeclared active substances were identified [11]. A large number of AASs can easily be obtained online, and their purchase is unregulated and anonymous, which makes the whole process of abuse easy. These are often substances of incompletely known content, of dubious quality and with insufficient information on their use and side effects [10,11].

There are two different ways to look at the analysis of growth stimulants. Either specific substances/groups of substances are searched for and analyzed in biological samples, or screening is carried out, where the aim is to identify the largest possible amounts of prohibited substances in the sample (e.g., food controls, doping controls). Community Reference Laboratories (CRLs) recommend analytical methods for national monitoring and control programs and have an accurate list of prohibited substances with acceptable analytical limits (MMPRs) in each type of sample matrix [12,13]. Resisting the pressure of new anabolic preparations on the market is becoming more and more difficult, which is why there is an effort to develop new and effective methods of detecting AASs, even in low concentrations and unknown chemical structures. Methods are beginning to focus on the identification of endogenous molecular biomarkers that are based on the physiological response induced by growth stimulants. Therefore, the effort is to understand the biological effects of AAS on the tissues of the organism as much and as comprehensively as possible. For that reason, in our experiment we focused on the selected parameters of pig muscle described below.

The basic parameters when observing the changes caused by AASs are muscle growth and, at the histological level, the growth of muscle fibers. Nowadays, a much-discussed parameter in the context of muscle growth is satellite cells. Muscle satellite cells are stem cells required for postnatal development and the regeneration of skeletal muscle, which retain the ability to both maintain a resting state in uninjured muscle and to be rapidly activated in response to growth or regeneration signals and resume the cell cycle [14,15,16]. They represent 2–11% of the sublaminal nuclei in muscle fibers and are located in a depression in the plasmalemma below the basal lamina of the muscle fiber [17,18]. Anabolic growth of skeletal muscles consists of several steps that essentially recapitulate the stages of development. First, muscle satellite cells must leave their quiescent state, become active and begin to proliferate. Asymmetric divisions are important for providing the myoblasts involved in the myogenic program as well as daughter cells that return to quiescence to add to the stem-cell pool. After proliferation, the myoblasts differentiate and fuse into myotubules, which get together with the original muscle fibers to repair them [17,19]. The behavior of satellite cells and their fate decisions are controlled by cell-autonomous mechanisms as well as external stimuli. However, recent studies have provided direct evidence of muscle regeneration and hypertrophy without any contribution from satellite cells, suggesting that satellite cells might be dispensable in some processes. Thus, a comprehensive picture of the interrelated mechanisms that control muscle stem-cell activity still needs to be defined [18,20,21,22].

The topic of the authors of the present study is very current all over the world, and therefore, more and more studies are dealing with the effects of AASs on muscle mass. However, most of the studies were conducted on animal models such as mice [23,24], rats [25,26,27,28], broilers [29] and rabbits [30] and were not focused on the abuse of AASs in fattening animals. A large number of studies are also devoted to the control of anabolic hormones in doping in sports where horses [31,32] or humans [33,34,35,36] are studied. There are studies on cattle [37,38,39] and sheep [40], although in many European countries, pork is consumed to a much greater extent than beef and lamb.

This study tests the hypothesis that AAS administration results in changes in porcine skeletal muscle at the histological level. The aim was to study the long-term effects of testosterone and nandrolone on changes in the diameter and area of muscle fibers, the amount of endomysium, the number of myonuclei and the number of muscle satellite cells. In this way, we aimed to determine the dependence of the histological parameters of pig muscle on the application of AASs and, thus, to determine potential supplementary markers for the detection of the presence of AAS in pork meat.

## 2. Materials and Methods

### 2.1. Ethical Statement

The experimental study, feeding and sampling were carried out at the Veterinary Research Institute in Brno, Czech Republic. The laboratory work was carried out at the Mendel University in Brno, Czech Republic. All experimental procedures were approved by the Central Commission for Animal Protection of the Czech Republic, serial number MZe 2081, no. 6931/2019-MZE-17214 on 7 February 2019.

### 2.2. Animals and Protocol

In total, 24 hybrid pigs of the OL 48 line (origin: Bioprodukt Knapovec Corp., Czech Republic), i.e., hybrids of Large White × Landrace (sow) × Duroc (boar), were used for the experiment. The pigs were divided into four groups of six based on the different applications and dosages of steroid drugs. Testosterone was administered to the first group—Sustanon preparation at 250 mg/mL (30 mg Testosterone propionas; 60 mg Testosterone phenylpropionas; 60 mg Testosterone isocaproas; 100 mg Testosterone decanoas propionate and a total amount of 17β-testosterone at 176 mg/mL, Organon, CZ, Reg. no. 56/357/91-C); nandrolone was administered to the second group—the veterinary preparation Myodine at 25 mg/mL (ester-laureate of 19nor-17beta-testosterone, Le Vet Beheer B.V., The Netherlands, CZ Reg. no. 96/030/17-C); a mixture of the previous two preparations was administered to the third group (Sustanon and Myodine in a ratio 1:1); and the fourth group was a control group without the application of anabolic steroid hormones. The groups were designated as the control, testosterone, nandrolone and mixed groups. In the first stage of the experiment, which lasted 92 days in total, 4 mg/kg b.w. testosterone i.m., 2 mg/kg b.w. nandrolone i.m. and 2 mg/kg b.w. mix i.m. were administered to two-month-old pigs to verify pharmacokinetics. In the second stage of the experiment, which lasted from day 93 to day 114, the amount of the administered substances was increased, in order to obtain positive samples of all tissues. Pigs were given 3 × 4 mg/kg b.w. of testosterone i.m., 3 × 5 mg/kg b.w. nandrolone i.m. and 3 × 5 mg/kg b.w. mix i.m., always after four days. The concentrations of AASs were selected based on the pharmacokinetic results of the preclinical trial before experiment. The pigs were slaughtered at approximately six months of age, after reaching slaughter weight (120–130 kg b.w.). The animal carcasses were handled according to applicable national legislation (Act. no. 266/1999 Coll.) and the legislation of European Union (Regulation (EC) No. 1069/2009).

### 2.3. Sampling

Muscle samples of a size of approximately 1 cm^3^ were taken from the *musculus longissimus dorsi* muscle from each animal immediately after killing. Samples for microscopy were fixed in 10% formaldehyde solution. Samples for immunofluorescence analysis were embedded in Tissue Tek cryoprotectant (O.C.T. Compound, Sakura Finetek, Torrance, CA, USA) and frozen in n-heptane placed on dry ice. Subsequently, the samples were frozen at −80 °C.

### 2.4. Light Microscopy

Muscle tissue samples were washed under running water after fixation for 24 h, dehydrated through an ascending ethanol series, cleared with xylene and paraffin embedded. Sections with a thickness of 5 μm (10 sections from each sample) were cut and stained for hematoxylin (Bamed, Litvinovice, Czech Republic) and eosin (Merck KGaA, Darmstadt, Germany). Histological preparations were observed on an Olympus BX51 microscope (Olympus, Tokyo, Japan) and scanned with a Promicam 3–5 CP camera (Promicra s.r.o., Praha, Czech Republic). Four parameters were evaluated by the QuickPHOTO MICRO 3.2. program (Promicra s.r.o., Czech Republic) for each section on ten muscle fibers: muscle fiber diameter, muscle fiber area, number of nuclei in muscle fibers. The amount of endomysium among muscle fibers was measured semiquantitatively and classified into two grades (+ and ++) according to the distance of the muscle fibers from each other.

### 2.5. Immunofluorescence Analysis of Satellite Cells

Slides (10 sections from each sample) with muscle tissue were removed from −80 °C. The tissue was carefully circumscribed with a PAP pen for immunostaining (Merck KGaA, Germany) for water repellency. Samples were overlaid with a Protein Block (Baria, Praha, Czech Republic) for 30 min at room temperature. Slides were incubated for 60 min in the dark with primary antibodies Anti-Hu CD56 (Exbio, Praha, Czech Republic, at a dilution of 1:100) to label satellite cells and Laminin Rat Monoclonal Antibody (OriGene Technologies, Rockville, MD, USA, at a dilution of 1:800) to label the basal line of muscle fibers. After incubation with primary antibodies, slides were washed in PBS + 0.1% Tween 20 (Merck KGaA, Germany) for 3 × 4 min. Appropriate secondary antibodies were applied—Goat anti-Rat IgG (H + L) Cross-Adsorbed Secondary Antibody, Alexa Fluor 488 (Thermo Fisher Scientific, Brno, Czech Republic, in a dilution of 1:1000) and Goat anti-Mouse IgG2a Cross-Adsorbed Secondary Antibody, Alexa Fluor 594 (Thermo Fisher Scientific, Czech Republic, in a dilution of 1:1000), and the slides were incubated for 60 min in the dark. After incubation, the slides were washed again for 3 × 4 min. To stain myonuclei, the slides were incubated for 10 min in the dark with DAPI (Merck KGaA, Germany). After the final washing step, all slides were mounted with ProLong Diamond Antifade Mountant (Thermo Fisher Scientific, Czech Republic) and folded with cover glass. After staining, digital images were taken using fluorescence microscopy on an Olympus BX51 microscope (Olympus, Tokyo, Japan) and scanned with a Promicam 3–5 CP camera (Promicra s.r.o., Czech Republic). Four parameters were determined: the number of muscle fibers, the diameter of muscle fibers, the number of satellite cells and the number of all cells (all nuclei). The fiber diameter was always measured for ten fibers from one section. The numbers of fibers, satellite cells and nuclei were always counted in the determined field of view on each section.

### 2.6. Statistical Analysis

First, the normality of the raw data distribution was tested using the Kolmogorov–Smirnov test. The normality distribution of the data was confirmed. The results were then analyzed using a two-sample *t*-test to determine significant sources of variability. Subsequently, a correlation analysis was performed between biologically related parameters using a linear regression model. Significance was determined for the differences between the control group and the experimental groups (testosterone, nandrolone and mixed) and for the mixed group and individual AASs (testosterone and nandrolone groups) with the following parameters: muscle fiber diameter, muscle fiber area, the number of muscle fibers, number of all cells (all nuclei) in muscle fibers and the number of satellite cells. GraphPad Prism ver. 8 software (GraphPad Software, La Jolla, CA, USA) was used for these statistical analyses. To confirm the hypotheses, a multivariate statistical analysis was performed using the PCA (Principal Component Analysis) method in the program Statistica v. 13.3 (DataBon s.r.o., Praha, Czech Republic).

## 3. Results

The results were obtained based on two measurement techniques: first, light microscopy and then immunofluorescence labeling.

### 3.1. Light Microscopy Analysis

The samples of the control group corresponded to the physiological and histological descriptions of the muscle. Round to oval compact muscle fibers separated by a small amount of intercellular mass were observed on the cross-section. Clearly visible nuclei were distributed around the fibers. Statistically significant differences were found between the control group and the groups that were administered testosterone, nandrolone and a mixture of testosterone and nandrolone. Under light microscopy, visible changes were observed in three monitored parameters—muscle fiber diameter, muscle fiber area and amount of endomysium between muscle fibers. The number of nuclei in muscle fibers did not prove to be significant. The parameters describing the size of muscle fibers showed statistically significant (*p* < 0.01) increases in the experimental groups with administered anabolic substances (Table 1).

Table 2 shows the comparisons between groups. Values from immunofluorescence labeling were measured to confirm the results. As differences were demonstrated between most groups, two separate tables were created for clarity.

A statistically significant difference (*p* < 0.01) was noted between the diameters of the muscle fibers of the control group and all experimental groups. In the testosterone group, there was an average 32% increase compared to the control; in the nandrolone group, there was a 49.7% increase; and in the mixed group, there was a 30.1% increase. Muscle fiber area was positively correlated with muscle fiber diameter. The experimental groups were increased in fiber area compared to the control; the testosterone group was increased in area by 112.2%, the nandrolone group by 145.7% and the mixed group by 107.3% (Figure 1). The low values of the standard error of the mean reflect the uniformity of the measured data. The amount of endomysium between muscle fibers was assessed semiquantitatively. In the control group, the amount was evaluated as two ++; the individual muscle fibers were surrounded by a larger layer of endomysium and were, thus, visibly separated from each other. In the experimental groups with administered AASs, the amount of endomysium was evaluated as one +; the muscle fibers were touching each other on most of the surface or very closely adjacent to each other. The amount of endomysium was therefore low, at approximately three times less than in the control group. The total number of nuclei in the muscle fibers showed a statistically significant difference (*p* < 0.01) between all groups except the control and nandrolone groups. From the results, we can suggest that this parameter is not suitable for evaluating the influence of AASs. For objective results and the true number of nuclei in a muscle fiber, the nuclei would have to be counted along the entire length of the muscle fiber and not just in the section, where the fibers may or may not be captured and the results may be distorted.

### 3.2. Immunofluorescence Analysis

Immunofluorescence analysis was primarily used for the detection of satellite cells with a specific antibody. Samples from immunofluorescence labeling were also used to determine correlations between the parameters to maintain the correctness of the results. The results of the immunofluorescence measurements corresponded to the previous measurements in the light microscope. Correlations between the measured values were calculated (Table 3). We focused on the differences between the control group and the experimental groups and on the differences between the single application of AASs and the mix.

A significant difference (*p* < 0.05) was noted between the muscle fiber diameters, with the testosterone, nandrolone and mixed groups increased in size compared to the control group. The number of fibers per unit area was negatively correlated with the muscle fiber diameter in all experimental groups, and the correlation coefficient was statistically significant (*p* < 0.05) in the testosterone and nandrolone groups (Figure 2). A statistically significant correlation coefficient (*p* < 0.05) also occurred in the testosterone group between fiber diameters and the number of all cells. The number of satellite cells increased by 123.85% in the testosterone group, by 213.30% in the nandrolone group and by 33.49% in the mixed group compared to the controls (Figure 3). In the nandrolone group, a positive correlation with a statistically significant correlation coefficient (*p* < 0.05) occurred between the number of satellite cells and the number of all cells.

### 3.3. PCA Analysis

The results of the PCA analysis of six measured parameters (fiber diameter from the optical microscope measurements, fiber diameter from the fluorescent microscope measurements, fiber area, fiber count, number of all cells and number of satellite cells) show that there was a statistically significant differentiation into individual clusters corresponding to individual groups, which confirms the above results. The two principal components (Factor 1 and Factor 2) explain 51.94% and 20.85%, which is altogether about 72%, of the variability in the data. Scatter plot PCA analysis showed a clear separation of the control group from the AAS-administered groups as well as differentiation between the experimental groups (Figure 4).

## 4. Discussion

Our aim was to study the influence of testosterone and nandrolone on changes in the structure of pig muscles. Both testosterone and nandrolone alone, as well as the mix of these two AASs, caused significant structural changes in the skeletal muscle of pigs, as we hypothesized. The most significant changes in the tissue were caused by the synthetic nandrolone and the mix of testosterone and nandrolone, with its effects closer to testosterone alone.

In this study, all the AASs used caused an increase in the diameter and area of the muscle fibers and a decrease in the amount of endomysium between these fibers. The reason is that AASs cause muscle hypertrophy, protein synthesis and metabolism of fat stores [10,11,12]. In general, the application of steroids leads to an increased growth rate of the fat-free mass. At the histological level, muscle growth can be noted as the increased diameter and area of muscle fibers and a significant increase in the proportion of fast low-oxidation fibers. Muscle fiber hypertrophy is accompanied by an increase in the number of myonuclei, with the primary source of the new myonuclei being activated and incorporated satellite cells otherwise located on the surface of the muscle fibers [23,24,25,26,27,33]. Most of the scientific works describing the effects of AASs on the histological parameters of muscle have been carried out on laboratory animals, such as mice [23,24], rats [25,26,27,28] and chickens [29]. Other studies were focused on control tests in sports doping, where the species were horse [31,32] or, in the non-veterinary area, a human [33,34,35,36]. For the effects of AASs on livestock, a few studies focus on cattle [37,38,39] and sheep [40], although pork has a much greater market impact in Europe. Most studies agree that AASs produce a measurable effect in muscle tissues and that the structural changes in the muscle tissue go beyond physiological values.

The results obtained from our experiment are very difficult to objectively compare with the results of other studies. Since identical hormonal preparations and their identical dosages were not used, AASs were applied at different intervals for different lengths of time, and in most cases, a different animal organism model was also used. The results are therefore comparable only in general terms, but they demonstrate that AASs increase the diameter and area of muscle fibers and reduce the amount of endomysium between muscle fibers. Regarding the other evaluated parameters, the number of nuclei and the number of satellite cells, the results of the studies differ considerably.

Most of the studies in which AASs were administered to experimental animals were conducted on rats, and they were mostly associated with physical exercise [26,41]. Studies agree that supraphysiological doses of steroids increase the muscle fiber diameter and surface area. The number of fibers per unit area was negatively correlated with the diameter of muscle fibers; increasing the diameter of the fibers resulted in a decrease in their number on the designated area, although more fibers were counted overall on the delineated surfaces of the experimental groups. These values could be explained by the decreasing amount of endomysium between the muscle fibers after the application of AASs, while the amount of endomysium in the experimental groups was semiquantitatively evaluated as three times lower than in the control. The fibers were therefore much closer to each other and, therefore, more could fit on a certain area. The difference in the number of nuclei in the muscle fibers did not appear as a statistically significant parameter in our study. As reported by Conceição et al. 2018 [33], the measurements were based on sections of muscle tissue. However, it is important to note that changes in muscle structure do not only occur in two dimensions. For the actual number of nuclei and the possibility of an objective assessment of this parameter, it would be necessary to count the number of nuclei along the entire length of the muscle fiber, not only on random sections, where the nucleus may or may not be captured. The conclusions of scientific articles differ in determining the number of nuclei in muscle fibers under the influence of AASs. A summary study by Conceição et al. 2018 [33] reported that muscle fiber hypertrophy of ≤10% already induces an increase in the number of nuclei in the fibers; a significant increase in the number of nuclei is then observed with greater muscle fiber hypertrophy (<22%) [33]. A study by Yu et al. 2014 [42] describes that the number of central nuclei in muscle fibers was very low and did not show a statistical difference between the control and test groups after the application of AASs. This could indicate that the increase in tissue as a result of anabolic action occurred due to the increase in the volume of muscle cells, i.e., hypertrophy, and not due to cell division. However, a study dealing with the long-term effects of steroid hormones on muscle tissue in athletes claims that long-term supplementation with anabolic steroids significantly increases both the total number of nuclei and the number of centrally stored nuclei [34]. This fact may indicate the activation of satellite cells and the conclusion that steroids can induce both muscle hypertrophy and hyperplasia.

Satellite cells (SCs) are crucial for the development, growth and maintenance of muscle tissue and are associated with skeletal muscle remodeling after their damage. After the application of AASs, there is a massive growth of muscle mass and hypertrophy of muscle fibers, and, therefore, we can assume a significant activation and proliferation of SCs in response to this condition. Our results show that after the application of AASs, there was an increase in the number of SCs on the surface of the muscle fibers; after the application of testosterone and nandrolone alone, the increase was up to multiples. In the nandrolone group, a positive correlation was found between the number of SCs and the number of all cells. However, the role of AASs in relation to muscle fiber cores and SCs is not clearly elucidated, and the results of studies differ widely here. The importance of SCs for hypertrophy after AAS application is described in chickens in the study of Asfour et al. 2021 [43] and in humans and others in Horwath et al. 2020 [44] and Machek et al. 2021 [45]. However, the studies clearly do not agree on the fact that the administration of AASs leads to increased numbers of SCs, as assumed by most of the original hypotheses. One of the mechanisms by which AASs regulate muscle growth is increased myogenic activity, i.e., the activation and subsequent fusion of SCs into new muscle fibers. In this, however, hypertrophy in response to a growth stimulus is associated with satellite cell-mediated myonuclear addition. A study by Yu et al. 2014 [42] talks about both muscle hypertrophy and hyperplasia associated with the use of AASs and associated with the activation of SCs and their fusion with existing muscle fibers, which leads to a higher number of myonuclei. However, some studies report that the observed increase in muscle fibers is not accompanied by the addition of new myonuclei, and the numerical increase can only be observed in SCs [43]. A study by Horwath et al. 2020 [44] reports an increase in the number of SCs up to more than 30%, while other studies report a statistically unchanged number of SCs after growth stimulation [17,18,19,20,21]. It appears that there might be a certain limit where muscle fibers are able to hypertrophy without the participation of the SCs, but this limit has not yet been established. Studies discuss the possible role of sex, species, time of hormone administration and other variables but do not reach a clear final conclusion [17,18,19,20,21,44].

In our study, in general, nandrolone, as a synthetic AAS, had the strongest effect on the muscle tissue. In the nandrolone group, a positive correlation with a statistically significant correlation coefficient was found between the number of all cells and the number of SCs on the surface of the muscle fibers. This supports the hypothesis of a mechanism of action of AASs that increases myogenic activity. In such a case, previously dormant SCs are activated, proliferate and subsequently fuse with the damaged muscle fiber. Although a significant increase in the number of SCs after AAS application was noted in all groups, a correlation with a statistically significant correlation coefficient was only found with the number of other nuclei in the nandrolone group, as mentioned above. In the testosterone group, a negative correlation was even observed, although it was statistically insignificant, i.e., the increased number of SCs was not accompanied by an increase in the number of nuclei inside the muscle fibers. This fact could support the idea that muscle fibers are capable of hypertrophy to a certain limit without activating SCs. After it is exceeded and for further growth, myonuclear addition is necessary. In the same way, a negative correlation was found in group T between the diameters of fibers and the number of all cells (statistically significant) and between the diameter of fibers and the number of SCs. The fibers, thus, increased in diameter without adding or multiplying cores on their circumference. In the nandrolone group, in contrast, the correlation between the diameter of the muscle fibers and the number of all cells was positive, although not statistically significant. Since the greatest hypertrophy recorded occurred after nandrolone administration, it is possible that in this case, a threshold was crossed and SCs were activated.

## 5. Conclusions

The results of the study confirm our hypothesis that regular application of AASs leads to changes in the skeletal muscle of pigs at the histological level. The application of AASs, specifically testosterone, nandrolone and their mix, for 114 days led to an increase in the diameter and area of muscle fibers and a decrease in the amount of endomysium, i.e., bringing the fibers closer together. There was also a significant increase in the number of SCs, which was positively correlated with an increase in the number of all nuclei in the muscle fibers in the nandrolone group. However, this correlation was not found in the other groups. These measurements could support the idea already mentioned in previous studies on other animal models [42,44] that muscle fibers are capable of hypertrophy without SC activation up to a certain limit. After the application of given doses of nandrolone, i.e., synthetic AASs, the threshold limit could be exceeded and the SCs activated.

This is the first study dealing with the effects of a combination of anabolic steroids on the histological parameters of pig muscle. We confirmed that there are certain dependencies of the histological parameters of pig muscle on the application of AASs. This information could help in the future for the development of additional screening methodologies for the detection of the presence of AASs in pork meat.

## Figures and Tables

**Figure 1 animals-13-02141-f001:**
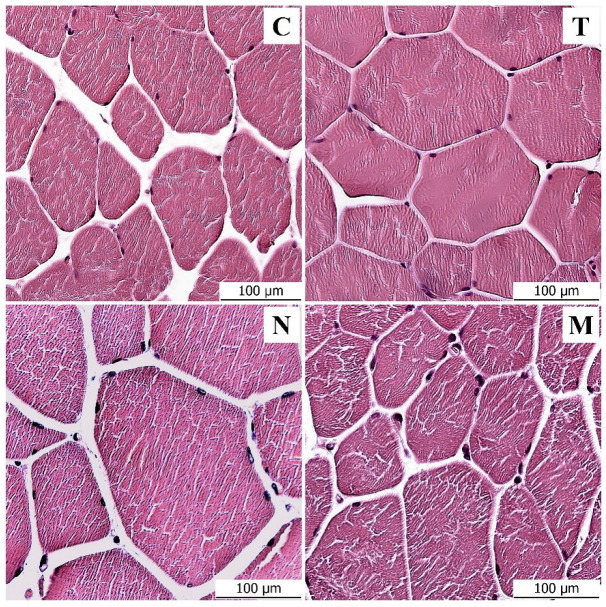
Histological staining of muscle fibers in the control group (C), testosterone group (T), nandrolone group (N) and group with a mix of testosterone and nandrolone (M). Scale bar = 100 μm.

**Figure 2 animals-13-02141-f002:**
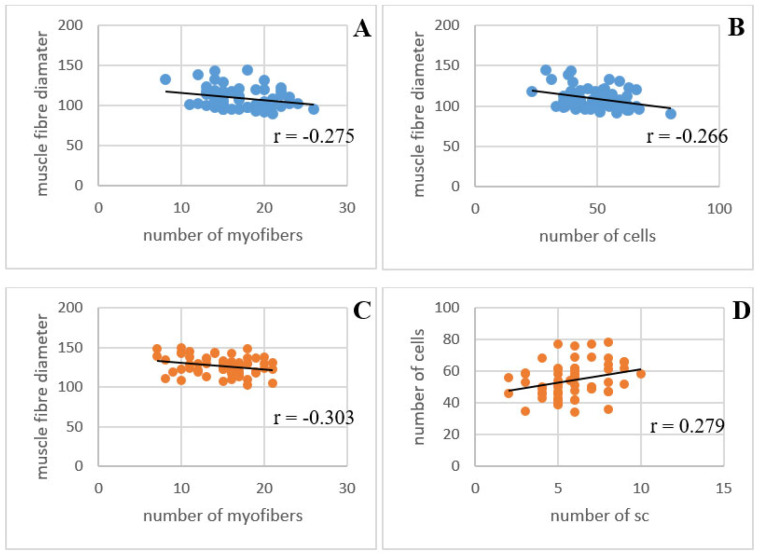
Pearson correlation analysis of measured values. Correlations in testosterone group (blue graphs) between number of fibers and muscle fiber diameter (**A**) and between number of all cells and muscle fiber diameter (**B**). Correlations in nandrolone group (orange graphs) between number of fibers and muscle fiber diameter (**C**) and between number of satellite cells and number of all cells (**D**), (*p* < 0.05).

**Figure 3 animals-13-02141-f003:**
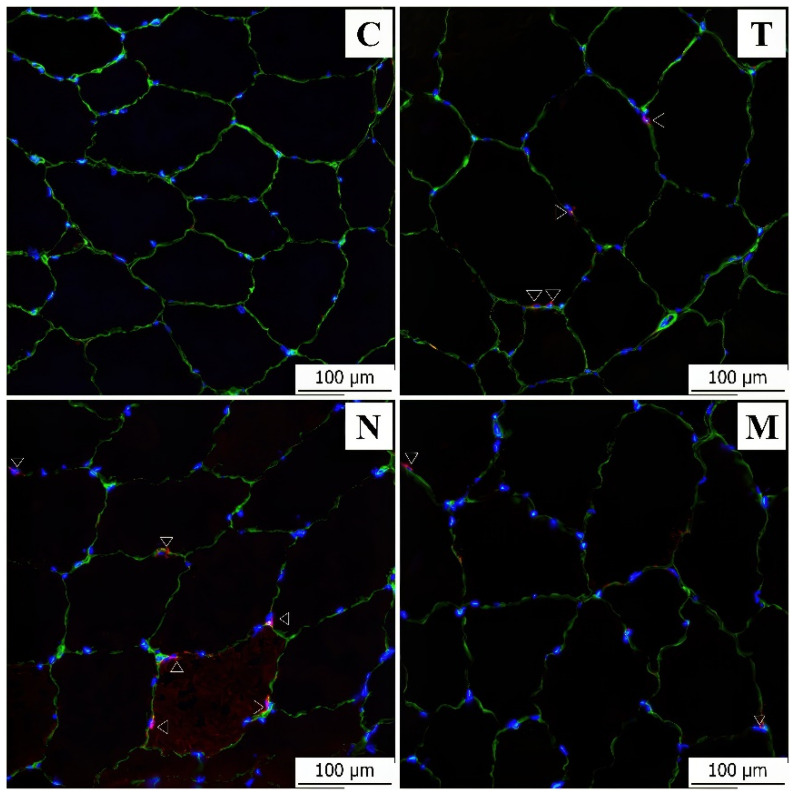
Immunofluorescence staining of satellite cells in the control group (C), testosterone group (T), nandrolone group (N) and group with a mix of testosterone and nandrolone (M). Sections were stained with an antibody against CD56 (red), laminin (green) and DAPI nuclear stain (blue). Arrowheads point to satellite cells. Scale bar = 100 μm.

**Figure 4 animals-13-02141-f004:**
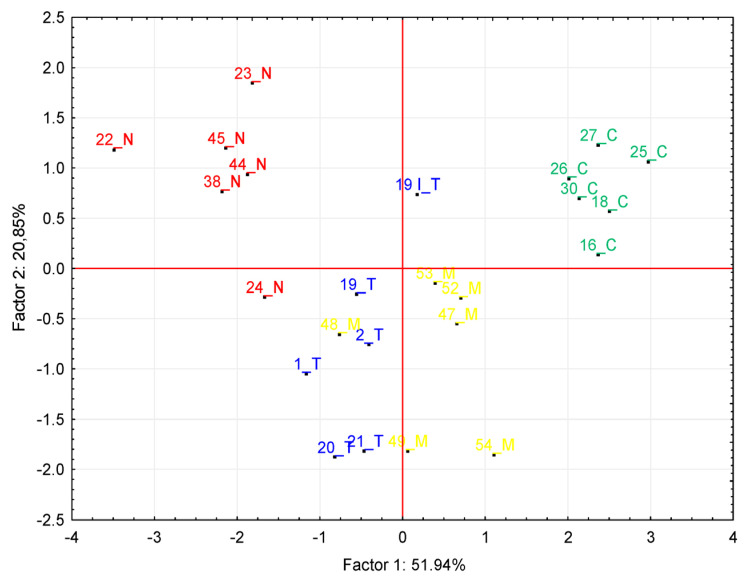
Scatter plots built with Factor 1 (x axis) and Factor 2 (y axis) for Principal Component Analysis (PCA) models calculated from measured values among the control group (C, green), testosterone group (T, blue), nandrolone group (N, red) and group with a mix of testosterone and nandrolone (M, yellow).

**Table 1 animals-13-02141-t001:** Average measured values of quantitative parameters ± SE.

	C	T	N	M
FD (μm)	72.27 ± 10.71	95.43 ** ± 14.48	108.20 ** ± 14.05	94.03 ** ± 10.41
FA (μm^2^)	3902 ± 931.60	8280 ** ± 2056	9589 ** ± 1949	8087 ** ± 1556
NN	5.12 ± 1.02	5.79 ** ± 1.32	5.03 ± 1.10	6.02 ** ± 1.88

Fiber diameter (FD), fiber area (FA), number of nuclei (NN). The groups were labeled as control (C), testosterone (T), nandrolone (N) and mix of testosterone and nandrolone (M). ** Statistically significant difference was related to the control group, *p* < 0.01.

**Table 2 animals-13-02141-t002:** Differences in average values of quantitative parameters of individual groups ± SE.

	C × T	C × N	C × M	T × N	T × M	N × M
FD (μm)	23.17 **	35.89 **	21.76 **	12.72 **	−1.4	−14.17 **
	±1.343	±1.317	±1.113	±1.504	±1.329	±1.303
FA (μm^2^)	4378 **	5688 **	4185 **	1309 **	−193	−1502 **
	±168.200	±161.000	±135.175	±211.200	±192.184	±185.887
NN	0.67 **	−0.09	0.90 **	−0.7611 **	0.23 **	0.99 **
	±0.125	±0.112	±0.159	±0.128	±0.171	±0.162

Fiber diameter (FD), fiber area (FA), number of nuclei (NN). The groups were labeled as control (C), testosterone (T), nandrolone (N) and mix of testosterone and nandrolone (M). An increase in values between groups is expressed as a positive value, and a decrease in measurements is expressed as a negative value. ** Statistically significant difference between compared groups, *p* < 0.01.

**Table 3 animals-13-02141-t003:** Correlation between measured values.

	C	T	N	M
FD × FC	0.059	−0.275 *	−0.266 *	−0.152
FD × NAC	0.024	−0.303 *	0.157	−0.190
FD × NSC	0.039	−0.196	−0.002	0.144
NSC × NAC	−0.036	−0.035	0.279 *	0.151

Fiber diameter (FD), fiber count (FC), number of all cells (NAC), number of satellite cells (NSC). The groups were labeled as control (C), testosterone (T), nandrolone (N) and mix of testosterone and nandrolone (M). * Statistically significant correlation coefficient (*p* < 0.05).

## Data Availability

The data presented in this study are available in the article. Further information is available upon request to the corresponding author.

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
