# Peer review of "Structural Changes in the Skeletal Muscle of Pigs after Long-Term Administration of Testosterone, Nandrolone and a Combination of the Two"

_animals, 2023, doi:10.3390/ani13132141_

Round 1
Reviewer 1 Report
The manuscript meets the scope of the journal and brings new insight into the detection of AAS in pork meat based on histological analyses of pig muscle fibers.
Title page: The title of the article is clear and informative.
The abstract is clear including all parts of traditionally accepted original article abstracts.
The introduction widely discusses the problem of the use of AAS for increasing muscle mass in meat-producing animals and shows this problem in European countries where pork is the main consumed meat.
Material and methods: The methods used are appropriate for obtaining significant data. L141: was instead of were; L154: 17β-19-nortestosterone; What are the usually used doses of AAS in practice? Why did you decide for the doses 3x higher in the 2nd experiment? Have you tested other doses before, e.g. 2x or 4x with similar results?; L169: Musculus with small m.
Results: Results are clear with reference to their tabular and graphical representation. L243-248: I suggest moving this part beginning with the second sentence to M&M; I suggest including statistical significances shown in Tables 2 and 3 in the text; Table 1: Mark significantly different values using asterisks and explain it in the note.; Tables 1 and 2: Use a decimal dot, not a comma; the same for the significant coefficient in L292 and L308; L320: 51.94%.
Discussion: This part is deeply discussed with the relevant global literature. L336: increased; L398: use only abbreviation SCs and explain it earlier at first mention in the L390; L407: talks; L413, 416: give a reference to “other studies” and “Studies discuss..”; L418-438: There is no reference for the comparison of your results?
References: References used are formally appropriate.
Reviewer 2 Report
The Authors have performed the interesting study aimed on the effects of the anabolic steroids administration on the muscle histology in pigs during the fattening period. The vast body of knowledge showed the effects of AAS administration in human and laboratory animal species. The illegal administration of AAS in fattening pigs is vast problem . In the USA and EU countries its administration to pigs is prohibited, thus there is an importance to track the effects and to find the potential markers of their administration in animals. According to the fact that the results of the similar studies are sparse the study is of a high novelty.
In the proposed paper the authors concentrated on the AAS administration on the muscle histology of pigs and clearly demonstrated the anabolic effects of those substances. The number of animals included in the experiment, animals treatment and material collection were approved by the adequate Ethical commission. The methods used for the evaluation of the effects of the AAS on the histological landscape are sufficient. However, I would like to ask why the authors have not analyzed expression of known proliferation markers in the Satellite Cells e.g. PCNA? The results of the study are clearly presented. Conclusions drawn from the study are also appropriate. Thus, I am recommending the publication of the proposed article in the Animals journal in the present form.
